# Combined Analysis of Metabolomics and Transcriptome Revealed the Effect of *Bacillus thuringiensis* on the 5th Instar Larvae of *Dendrolimus kikuchii Matsumura*

**DOI:** 10.3390/ijms252111823

**Published:** 2024-11-04

**Authors:** Jinyan Li, Qiang Guo, Bin Yang, Jielong Zhou

**Affiliations:** 1College of Biological Science and Food Engineering, Southwest Forestry University, Kunming 650224, China; lijinyan@swfu.edu.cn (J.L.); guoqiang@swfu.edu.cn (Q.G.); 2Key Laboratory of Forest Disaster Warning and Control of Yunnan Province, Southwest Forestry University, Kunming 650224, China

**Keywords:** *Dendrolimus kikuchii Matsumura*, *Bacillus thuringiensis*, biochemical, pathological characteristics, transcriptome, metabolomics

## Abstract

*Dendrolimus kikuchii Matsumura* (*D. kikuchii*) is a serious pest of coniferous trees. *Bacillus thuringiensis* (*Bt*) has been widely studied and applied as a biological control agent for a variety of pests. Here, we found that the mortality rate of *D. kikuchii* larvae after being fed *Bt* reached 95.33% at 24 h; the midgut membrane tissue was ulcerated and liquefied, the MDA content in the midgut tissue decreased and the SOD, CAT and GPx enzyme activities increased, indicating that *Bt* has toxic effects on *D. kikuchii* larvae. In addition, transmission electron microscopy showed that *Bt* infection caused severe deformation of the nucleus of the midgut tissue of *D. kikuchii* larvae, vacuoles in the nucleolus, swelling and shedding of microvilli, severe degradation of mitochondria and endoplasmic reticulum and decreased number. Surprisingly, metabolomics and transcriptome association analysis revealed that four metabolic-related signaling pathways, Nicotinate and nicotinamide metabolism, Longevity regulating pathway—worm, Vitamin digestion and absorption and Lysine degradation, were co-annotated in larvae. More surprisingly, Niacinamide was a common differential metabolite in the first three signaling pathways, and both Niacinamide and L-2-Aminoadipic acid were reduced. The differentially expressed genes involved in the four signaling pathways, including *NNT*, *ALDH*, *PNLIP*, *SETMAR*, *GST* and *RNASEK*, were significantly down-regulated, but only *SLC23A1* gene expression was up-regulated. Our results illustrate the effects of *Bt* on the 5th instar larvae of *D. kikuchii* at the tissue, cell and molecular levels, and provide theoretical support for the study of *Bt* as a new biological control agent for *D. kikuchii*.

## 1. Introduction

Conifer trees play an important role in the ecosystem, and they not only make important contributions to ecological balance and biodiversity, but also have high economic value as an important timber resource. However, conifers are threatened by *Dendrolimus kikuchii Matsumura* (*D. kikuchii*). *D. kikuchii* belongs to the genus Dendrolimus of Lepidoptera, and it is a leaf-eating pest that often occurs on coniferous trees such as *Pinus kesiya var. langbianensis*, *Pinus yunnanensis*, *Pinus massoniana Lamb.* and *Keteleeria evelyniana Mast.* [1,2,3]. It is concentrated in Yunnan province, Hunan province, Sichuan province, Guangdong province and other southern regions in China (Figure 1A) [4]. The larval stage of *D. kikuchii* is divided into 1–7 instars (Figure 1B). It enters the gluttony period from the 4th instar, with increased body size, larger feeding amount and longer feeding time [5,6]. When the pest occurs seriously, it will not only lead to serious ecological damage and economic losses, but also cause people to suffer from pine caterpillar disease after the toxic hair of the caterpillar contacts the human body, which can cause inconvenience to human life [7,8]. Biological control has made great progress in recent years. Compared with chemical control [9,10], biological control has the advantages of good effect, low cost and low environmental pollution. Therefore, in order to effectively control *D. kikuchii* and avoid the high toxicity of pesticides and the resulting health and environmental risks [11], it is urgent to find a biological control agent that can replace pesticide insecticides.

Many bacteria have become an effective choice for pest control because they can produce protein which is toxic to insects [12]. *Bacillus thuringiensis* (*Bt*) is the most widely used biological control agent for insect-pathogenic bacteria. *Bt* is a Gram-positive rod-shaped bacterium, belonging to the Bacillus family, which has the ability to form resistant spores [13]. When *Bt* cells sporulate, they form a parasporal Crystal composed of protein, which contains many kinds of insecticidal proteins (Δ-endotoxin, also known as Cry and Cyt toxin) encoded by the Cry gene and Cyt gene [14,15]. When insects feed on *Bt*-infected food, parasporal crystals release prototoxin in the insect midgut, which is hydrolyzed by protease to become active toxin. The active toxin interacts with insect midgut epithelium, leading to the destruction of midgut tissue cells, which in turn affects the integrity of the midgut membrane, causes inflammation and other diseases and finally leads to insect death [16,17,18]. It is precisely because of the toxic activity of the Cry protein of *Bt* to insect larvae that these toxins can be used for biological control of pests by spraying formula and transgenic crops, including Cry protein or some active fragments [19,20,21]. It is reported that *Bt* is active against pests such as Lepidoptera [10], Diptera [12], Coleoptera [22,23] and Hymenoptera [24,25], but it is safe for humans, aquatic organisms, bees and other organisms to use at appropriate concentrations [11]. Therefore, *Bt* has great potential as a biological control agent for *D. kikuchii*.

Compared with other pests, there are few studies on *D. kikuchii*, especially on its metabolome and transcriptome. In addition, as far as we know, the effect of *Bt* on *D. kikuchii* has not been reported so far. In this study, we used *Bt* to infect the 5th instar larvae of *D. kikuchii*, and observed and analyzed the poisoning phenomenon of larvae after different infection times from the aspects of mortality, anatomical observation and biochemistry. At the same time, for the first time, we used transmission electron microscopy to analyze the midgut tissue of *D. kikuchii* larvae infected with *Bt* at different times. Finally, the effects of *Bt* at the gene transcription level and protein translation level of *D. kikuchii* larvae were revealed by the transcriptome and metabolome and their combined analysis. Our research will provide theoretical support for *Bt* as a biological control agent for *D. kikuchii*.

## 2. Results

### 2.1. Bt Has Toxic Effect on the 5th Instar Larvae of D. kikuchii

Although *Bt* has been proved to be toxic to many insects, including Lepidoptera, Coleoptera and Diptera, similar research on *D. kikuchii*, which also belongs to Lepidoptera, has not been reported. Therefore, in order to explore whether *Bt* infection has toxic effects on *D. kikuchii*, we observed the larvae after infection and recorded and calculated the mortality rate. The 5th instar larvae of *D. kikuchii* are collectively referred to as larvae. The results showed that the mortality rate of the CK group larvae fed with PBS solution was very low in each time period, there was only one larva that died at the most and there was no difference in the mortality rate within and between groups at the four time nodes. For the *Bt* group larvae fed with *Bt* liquid, the average mortality rate was 2.22% at *Bt*0h, and the mortality rate reached 95.33% at *Bt*24h. At the same time, there was no difference in the mortality rate within the group at the four infection time nodes, but there was a significant difference in the mortality rate between the groups at each time node (Figure 2A). This indicated that *Bt* had an obvious toxic effect on *D. kikuchii*, and the toxicity was stronger with the extension of infection time.

### 2.2. With Prolongation of Bt Infection Time, the Midgut Membrane Tissue of D. kikuchii Langbianensis Festered and Liquefied

The midgut tissue is the digestive organ of insects and the most important immune organ. In order to confirm whether *Bt* infection causes damage to the complete structure of the midgut tissue, we dissected and observed the midgut tissue of larvae at different time points after *Bt* infection. The results showed that the midgut tissue and contents of *Bt*0h larvae were light green. The midgut tissue of *Bt*6h larvae was dark green; with the extension of *Bt* infection time, the color of midgut tissue gradually changed from dark green to yellow at *Bt*12h, and the lower end of midgut tissue had liquefied, without forming a complete midgut membrane structure, and the rest of the midgut tissue was also accompanied by the phenomenon of intestinal wall thinning and gradual liquefaction. At *Bt*24h, the color of the midgut tissue of the larvae was yellowish green, the structure had been completely damaged and all were liquefied (Figure 2B).

### 2.3. After Bt Infection, MDA Content Decreased, CAT, SOD and GPx Enzyme Activities Increased

In the face of invasion by exogenous microorganisms, the midgut tissue of insects is damaged and produces toxic substances, which stimulates the immune defense mechanism of insects and enhances a series of detoxification enzyme activities, thus eliminating toxic substances and eliminating pathogenic microorganisms so as to maintain their own steady-state balance and reduce the damage to their body. We measured the MDA content and CAT, SOD and GPx activities in the intestinal contents of the larval midgut tissue, and found that the MDA content showed a trend of decreasing–increasing–decreasing, which decreased at 6 h and 24 h after infection, respectively, and was significantly different from that at *Bt*0h, but the MDA content at *Bt*6h, *Bt*12h and *Bt*24h was lower than that at *Bt*0h. Compared with *Bt*0h, the activities of CAT, SOD and GPx increased at 6 h, 12 h and 24 h after *Bt* infection, and decreased after reaching the highest value at 6h (Figure 2C).

### 2.4. The Analysis of Pathological Characteristics Showed That the Cell Structure of Midgut Tissue of Larvae Was Damaged After Bt Infection

The previous results have shown that the midgut tissue of *Bt*-infected larvae is destroyed. Therefore, in order to further explore the effect of *Bt* on the midgut tissue of *D. kikuchii,* we used transmission electron microscopy to observe the midgut tissue in detail. In the *Bt*0h group, we observed that the nuclear morphology of epithelial cells in midgut tissue was full, the nuclear membrane structure was complete and the substances in the nucleus were evenly distributed. In the *Bt*6h group, the nucleus of the epithelial cells began to elongate and deform, the gap between the inner membrane and the outer membrane of the nuclear membrane widened and the substances in the nucleus aggregated. In the *Bt*12h group, the shape of the epithelial cell nucleus obviously changed, with irregular shape, vacuoles in the nucleolar center, a protruding nuclear membrane and angular shape. By the time of *Bt*24h, vacuoles had appeared in the nuclear substance (Figure 3A–D). For mitochondria, we observed that in *Bt*0h group, the double-layer membrane structure of mitochondria was complete, the inner ridge was clearly visible, and most mitochondria were normal oval or spindle-shaped. After *Bt* infection, the morphology of mitochondria was elongated and deformed, vacuoles appeared at 6 h and the inner ridge was blurred. After 12 h, the deformation degree of mitochondria deepened, vacuoles appeared in the middle and the number decreased (Figure 4A–D). In the *Bt*0h group, the endoplasmic reticulum was arranged in an orderly manner and in large numbers. After 6 h of *Bt* infection, the endoplasmic reticulum slightly expanded, deformed and twisted. At *Bt*12h, the plasmodesmata was deformed and expanded seriously, arranged sparsely, began to degrade and the number decreased. The endoplasmic reticulum was severely degraded at *Bt*24h, and the number was significantly reduced compared with the *Bt*0h group (Figure 5A–D). In addition, the microvilli in the *Bt*0h group was developed well and arranged closely. At *Bt*6h, the microvilli slightly swelled; At *Bt*12h, the deformation, fracture and shedding of microvilli are gradually aggravated, and the arrangement is chaotic and the fracture is serious. At *Bt*24h, it was observed that the microvilli fell off and broke seriously (Figure 6A–D).

### 2.5. Screening of Differential Metabolites and KEGG Enrichment Analysis

Mortality analysis, anatomical observation and pathological observation all showed that *Bt* had toxic effect on the 5th instar larvae of *D. kikuchii*. Cell biochemical determination showed that the MDA content and CAT, SOD and GPx activities in the larvae changed after *Bt* infection, indicating that the metabolism of the larvae changed after *Bt* infection. Therefore, in order to further study the influence of *Bt* infection on *D. kikuchii* at the metabolic level, we further measured the metabolic spectrum of midgut tissue. A total of 411 DEMs were obtained, and 292 DEMs were obtained in the *Bt*6h vs. the *Bt*0h group, of which 52 were up-regulated and 240 were down-regulated (Figure 7A). A total of 97 DEMs were obtained in the *Bt*12h vs. the *Bt*0h group, of which 22 were up-regulated and 75 were down-regulated (Figure 7B). A total of 165 DEMs were obtained in the *Bt*24h vs. the *Bt*0h group, of which 53 were up-regulated and 112 were down-regulated (Figure 7C). The differential metabolites of three groups in *D. kikuchii* were intersected, and a total of 27 differential metabolites were obtained (Figure 7D).

By matching with the database of human metabolites, 27 differential metabolites were divided into six categories (Figure 7E), among which organic acids and derivatives accounted for the most, including 8 differential metabolites. They are Thiodiacetic acid, Cystathionine ketimine, S-(3-OXO-3-Carboxy-N-Propyl) Cysteine, L-2-Aminoadipic acid, Gamma-Glutamyltyrosine, Gamma-Glutamylphenylalanine, Gamma-Glutamyltryptophan and Nacetylproline. KEGG enrichment analysis results show that, Longevity regulating pathway—worm, Penicillin and cephalosporin biosynthesis, Riboflavin metabolism, Lysine biosynthesis, Vitamin digestion and absorption, Lysine degradation and Nicotinate and nicotinamide metabolism were significantly enriched (Figure 7F). Topological analysis shows that Nicotinate and nicotinamide metabolism signal pathways are the key signal pathways, and nicotinamide, a differential metabolite, has the greatest influence weight in the pathway (Figure 7G).

### 2.6. Transcriptome Sequencing Results Analysis

We sequenced the transcriptome of midgut tissue of 5th instar larvae of *D. kikuchii* after different infection time. The results showed that 3446 DEGs were screened according to p adjust < 0.05 and |log2FC| ≥ 1. A total of 1261 differentially expressed genes were obtained in the *Bt*6h vs. the *Bt*0h group, of which 615 were up-regulated and 646 were down-regulated (Figure 8A). A total of 1446 DEGs were obtained in the *Bt*12h vs. the *Bt*0h group, of which 802 were up-regulated and 644 were down-regulated (Figure 8B). A total of 2276 DEGs were obtained in the *Bt*24h vs. the *Bt*0h group, of which 1073 were up-regulated and 1203 were down-regulated (Figure 8C). The results showed that after 6 h and 12 h of *Bt* infection, the number of differential genes in the two groups was similar. Compared with 6 h and 12 h, after 24 h of infection, the number of differential genes was obviously more than the first two. Venn analysis was performed on the total DEGs of the three groups, and a total of 388 intersecting genes were obtained (Figure 8D).

In order to further understand the functional information of 388 genes, we conducted KEGG annotation analysis on them. A total of 225 signal paths were annotated. Coincidentally, the Longevity regulating pathway—worm signal pathway, Lysine degradation signal pathway, Nicotinate and nicotinamide metabolism signal pathway and Vitamin digestion and absorption signal pathway also appeared in 27 KEGG enrichment results of differential metabolism.

### 2.7. Correlation Analysis of Transcriptome and Metabolomics

In order to further confirm the accuracy of the results of metabolism level and transcription level, we analyzed the correlation between transcription and metabolism of 388 differentially expressed genes and 27 differentially expressed metabolites. KEGG enrichment analysis showed that there was no common enrichment pathway between the transcription and metabolism. Therefore, KEGG annotation analysis was carried out on them, and the results showed that seven signal paths were annotated together: Metabolism of xenobiotics by cytochrome P450, Biosynthesis of cofactors, Pentose and glucuronate interconversions, Nicotinate and nicotinamide metabolism, Lysine degradation, longevity regulating pathway—worm and Vitamin digestion and absorption (Figure 8E). It is worth noting that the latter four factors exist in KEGG enrichment, KEGG annotation and transcriptional metabolism correlation analyses of differential metabolites of differentially expressed genes. The four signal pathways and their genes and metabolites were sorted out. Niacinamide exists in all three pathways, and L-2-Aminoadipic acid is a differential metabolite in the Lysine degradation signaling pathway, and both of them are down-regulated. Among the seven differentially expressed genes, SLC23A1 was up-regulated, while the other six genes were down-regulated (Table 1).

### 2.8. RT-qPCR Verification

RT-qPCR verified the expression status of seven genes, and the relative mRNA expression of seven genes was consistent with the results of transcriptome sequencing. Compared with the Bt0h group, the expression levels of *NNT*, *PNLIP*, *SETMAR*, *GST* and *ALDH* decreased significantly at *Bt*6h, and then decreased gradually. *SLC23A1* increased significantly at *Bt*6h, and the expression of *SLC23A1* continued to increase with time. The results showed that the expression of the above seven genes was affected by *Bt* infection of *D. kikuchii* (Figure 9).

## 3. Discussion

*Bt* has become the most widely used biological insecticide because of its good insecticidal effect, strong specificity, lack of residue and safety to non-target organisms. It has toxic effects on insects such as Lepidoptera, Diptera, Coleoptera, Hymenoptera, Hemiptera, Orthoptera and poultry lice, and even on snails, nematodes and protozoa [26]. *Cotton boll weevil* (*CBW*) is a pest that harms cotton and causes huge economic losses to the American cotton industry [27]. Diego et al. [28] found a new *Bt* strain named *Bt*_UNVM-84, and its spore–crystal mixture had 91% insecticidal activity against the newly hatched larvae of *CBW*. *Dioryctria abietella* (*DA*), belonging to Lepidoptera and Coccinellidae, is a forest pest, which seriously harms Korean pine cones [29]. Wang et al. [30] applied a *Bacillus thuringiensis* subsp. *Nizawa* to the larvae of *DA*, and the mortality rate was as high as 100%. In this study, the mortality of the 5th instar larvae of *Dendrolimus kikuchii* treated with *Bt* was compared with that of the CK group. The mortality of the larvae in the *Bt* group increased with the prolongation of the infection time, and the difference was significant. The mortality rate reached 95.33% after feeding for 24 h, indicating that *Bt* had a strong toxic effect on the larvae of *D. kikuchii*.

The insect midgut is an important place to secrete digestive enzymes, digest food and absorb nutrients, and it is also the immune tissue of insects. It is considered that the target site of most *Bt* proteins is insect midgut epithelial cells. When the structure of insect midgut epithelial cells is destroyed, midgut tissue is damaged, resulting in inflammation and insect death [31]. Studies have shown that the midgut tissue of Lepidoptera plays an important role in the feeding process and is the main tissue affected by *Bt* toxin [32]. Many scholars have observed and analyzed the pathological changes in insect midgut tissue by electron microscopy. Zhang et al. [33] observed the pathological changes of midgut tissue of *Helicoverpa armigera* (*HA*) after feeding on Vip3Aa protein by electron microscope; Song et al. [34] observed the paraffin sections of the midgut of *Scarabaga aeruginosa* (*SA*) infected with *Bt* HBF-1 strain and that of *HA* infected with *Bt* Goldstein subspecies. After *HA* fed on HaHR3 transgenic tobacco, midgut cells undergo shedding of microvilli, nuclear degradation, nuclear membrane deformation, cell body cavity formation, mitochondria swelling and endoplasmic reticulum deformation [35]. After feeding on *Bt* strain, the midgut microvilli swelled, and bubble-like structures appeared inside, which became sparse and completely disappeared locally. The endoplasmic reticulum expands, breaks and the number of reticula decreases [31]. After infection by *Bt*05041, the midgut microvilli of *DA* were destroyed, the organelles were deformed and the cytoplasm was vacuolated. With the extension of treatment time, the connection between the basement membrane and diaphragm gradually breaks down until the cells are completely dissolved [36]. All the above studies show that midgut tissue is the main damaged part of insects infected by exogenous bacteria. In this study, with the extension of time after *Bt* feeding, the midgut tissue of the 5th instar larvae showed the characteristics of light green–dark green-yellow–yellow green, and the intestinal wall tissue showed the characteristics of normal–thin–fester–liquefaction. The results of transmission electron microscopy are similar to those of previous studies. With the extension of time after eating *Bt*, the midgut tissue cells show serious nuclear deformation, vacuoles in nucleoli, swelling and shedding of microvilli, serious degradation of mitochondria and endoplasmic reticulum and decreased number. The results showed that *Bt* entered the midgut tissue and destroyed the morphology, structure, quantity and even function of the organelles in the midgut tissue of the 5th instar larvae, which caused the midgut tissue to be damaged and cracked, so that the 5th instar larvae of *D. kikuchii* died.

*Bt* infection destroys the stability mechanism in insects, affects the transcription and metabolism of pests and forces the transcription and metabolism of insects to change. Huynh et al. [37] analyzed the metabolomics of *western corn rootworm* (*WCR*) larvae feeding on corn expressing *Bt* toxin and corn not expressing *Bt* toxin, and found that 724 metabolites had different changes. Dhania et al. [38] monitored the expression pattern of midgut genes of *Achaea janata* larvae exposed to sublethal doses of *Bt*. The results showed that 5002 genes were expressed in midgut tissues of *Achaea janata* larvae after eating *Bt*, including 1611 differentially expressed genes, 701 of which were up-regulated and 910 down-regulated. Jin et al. [39] found that *Bt* has a significant effect on many signal pathways of insects through the integrated analysis of transcriptomics and protein genomics. In this study, 27 intersection differential metabolites and 388 intersection differential expression genes were obtained. Seven signal pathways analyzed by the correlation between transcriptome and metabolomics are all related to metabolism, and Nicotinate and nicotinamide signal pathways and nicotinamide are the most important signal pathways and metabolites. Although the results of metabolomics and transcriptomics in this study are similar to those of previous studies, there are also differences, indicating that *Bt* infection does affect the transcription and metabolism of the 5th instar larvae of *D. kikuchii*, but the affected signal pathways, genes and metabolites will be different among different species.

MDA is a toxic substance produced by lipid peroxidation in mitochondria, which can stimulate and induce oxidative stress in the body and cause damage to the body. The elimination of toxic substances is mainly played by a series of detoxification enzymes, such as superoxide dismutase, catalase, glutathione peroxidase, etc. [40,41], and their gene expression or enzyme activity will be induced by the change in toxic substances. In this study, MDA content showed a trend of decreasing–increasing–decreasing, but it was always lower than the *Bt*0h group. Interestingly, the enzyme activities of the three enzymes have an opposite trend with MDA content. Compared with the *Bt*0h group, CAT, SOD and GPx all reached the highest activity at *Bt*6h, decreased at *Bt*12h and increased at *Bt*24h, with SOD decreasing continuously at *Bt*12h and *Bt*24h, but their enzyme activities in the *Bt* group were higher than those in *Bt*0h group. The results showed that *Bt* infection damaged the midgut tissue of *D. kikuchii* larvae, increased the MDA content and induced the activities of CAT, SOD and GPx to increase, which played an immune defense role and reduced the body damage. Aldehyde dehydrogenase (*ALDH*) is an oxidase with NAD+ as coenzyme and aldehydes as substrate, which can transform aldehydes into carboxylic acids and reduce the harm of toxic substances to the body. It also plays an important role in the process of MDA elimination. Studies have shown that *ALDH* can metabolize aldehydes produced by membrane lipid peroxidation [42]. Garcia et al. [43] found that the activity of acetaldehyde dehydrogenase in fish liver exposed to 0.01 mL/L and 0.1 ml/L biodiesel increased by 2.7 times and 3.9 times, respectively, and was negatively correlated with MDA level. However, after vitamin E deficiency induced lipid peroxidation and increased MDA content in mouse liver tissue, it was found that the increase in MDA content did not induce the change in *ALDH* activity [44]. In this study, the expression of *ALDH* continued to decrease, and the MDA content did not increase with the decrease in *ALDH*. We speculate that there are two reasons for this phenomenon: first, there is a feedback mechanism between MDA content and *ALDH* expression. When the highly active detoxification enzyme reduces the MDA content in cells, the *ALDH* expression using MDA as substrate also decreases. Secondly, some toxin released by *Bt* may inhibit the expression of *ALDH*, which makes *ALDH* not actively participate in MDA metabolism, and the other two enzymes play a compensatory role.

Nicotinamide adenine dinucleotide (NAD+) and nicotinamide adenine dinucleotide phosphate (NADP+) are soluble electron carriers, which are oxidized and reduced during the catabolism of energy substrates in cell fluids and mitochondria [45]. NAD+ and NADP+ are coenzymes of many antioxidant enzymes, including *ALDH*, CAT, *SOD*, etc. As specific electron donors, they participate in different metabolic pathways. In the mitochondria of many organisms, the redox states of NAD+ and NADP+ are interrelated. In order to maintain normal activities in the body, the ratio of NAD+ and NADP+ must be maintained in a relatively stable state. Amide nucleotide transhydrogenase (*NNT*) is a nuclear-encoded mitochondrial inner membrane protein, which has the function of redox-driven proton pump. *NNT* catalyzes the hydrogen transfer between NADH and NADP+, and uses proton gradient to make H+ move down along the electrochemical potential of mitochondrial inner membrane. The positive reaction of *NNT* leads to the ratio of NADPH/NADP+ in mitochondrial matrix becoming much higher than that of NADH/NAD+ [46], which makes *NNT* a prominent redox regulatory protein. Under the physiological conditions of mitochondria, *NNT* is an effective and important producer of NADPH. *NNT* can reduce one of the coenzymes at the expense of the oxidation of the other coenzyme by catalyzing the transfer of redox potential between the two coenzymes [45]. At present, there are many studies on the mutation and deletion of *NNT* gene in mice. The level of NADPH in the mitochondria of mice with *NNT* deletion decreased significantly, and the balance of NAD+ increased significantly, which led to the increase in superoxide and hydrogen peroxide [47]. *NNT* is also related to immune defense. It is worth noting that high NADPH levels may slow down the positive reaction of *NNT* [48]. In this study, nicotinamide decreased as a synthetic substrate of NADP+ and NAD+, which indicated that after being infected by *Bt*, a large number of detoxification enzymes and/or antioxidant enzymes were induced and activated in *D. kikuchii* larvae, which reduced the consumption of nicotinamide and synthesized two coenzymes to participate in the antioxidant reaction. As the invertase of NADP and NAD, the expression of the *NNT* gene decreased, presumably because the transcription of *NNT* gene was inhibited by the toxic protein released by *Bt*, which was a method for *Bt* to inhibit the antioxidant effect of insects, thus poisoning insects.

In addition to the antioxidant enzymes mentioned above, vitamin C is also an important antioxidant enzyme in the body. Vit C is a highly water-soluble vitamin, which exists in the body in two forms: reduced Vit C, namely L-ascorbic acid (ASC), and oxidized Vit C, namely L-dehydroascorbic acid (DHA), the former being dominant. ASC can be used as a natural reducing agent to remove ROS produced during cell metabolism. In many cells, DHA will be reduced to ASC in a series of enzymatic reactions. At this time, NADH acts as a monodehydroascorbic acid reductase to promote the transformation of DHA to ASC. *SLC23A1* is an L-ascorbate transferase, which depends on the electrochemical gradient of Na+ inside and outside the cell and mediates the synergistic and interactive transport of Na+ and ASC. In this study, the *SLC23A1* gene is the only gene whose expression is up-regulated, which indicates that infection leads to an antioxidant reaction in cells, which induces the up-regulation of *SLC23A1* expression so as to transport a large amount of L-ascorbic acid into cells to remove ROS. In addition, *PNLIP*, *GST*, *SETMAR* and *RNASEK* are also genes with great influence weight in this study. This study shows that the expression of these genes will be affected after being infected by *Bt* protein or *Bt* [49,50]. L-2-Aminoadipic acid is the final product of the Lysine degradation signal pathway, and the only signal pathway after synthesis to participate in the biosynthesis of penicillin and cephalosporin as a substrate, which is the specific precursor of all penicillin and cephalosporin [51]. Penicillin and cephalosporin are important antibiotics with bactericidal and bacteriostatic effects.

## 4. Materials and Methods

### 4.1. Dendrolimus kikuchii Matsumura

Collecting the cocoon of *D. kikuchii* (Taxodium yunnanense forest, Shuangcun village, Anning city, Yunnan province, 103 25′ E, 25 47′ N) comprised eclosion in the laboratory, mating between males and females, and collecting eggs. According to the method of Nie et al. [8], the larvae were raised in a cage indoors, with a feeding temperature of 28 ± 1 °C, relative humidity of 73 ± 2% and illumination period 8L:16D [52]. The Taxodium *Keteleeria evelyniana Mast.* was replaced every 2–3 days to ensure the cleanliness of the insect cage. Larvae developed to the 4th instar and entered the gluttony stage. However, since the larvae were too small to be sampled at this stage, the 5th instar larvae were selected as experimental insects in this study. When the larvae of *D. kikuchii* grew to the first day of the 5th instar, healthy larvae with similar body size were selected as experimental insects.

### 4.2. Bacillus thuringiensis

*Bt* was purchased from the China Agricultural Microbiology Collection and Management Center, with the strain type of *Bacillus thuringiensis* subsp. dendrolimus and the strain number of ACCC10062. According to the results of preliminary toxicity testing, the control group (CK) was provided PBS solution with a volume equal to the *Bt* bacteria solution, and the *Bt* concentration in the experimental group (*Bt*) was LC_50_ = 3.3 × 10^8^ spores mL^−1^.

### 4.3. Mortality Analysis and Anatomical Observation of Midgut Tissue

The CK group and the *Bt* group were set up with three replicate groups with 30 larvae in each replicate group. Fresh pine needles were treated by the dipping method [53], and the pine caterpillars were then allowed to feed freely, and the feeding conditions were observed and recorded. The mortality of CK group and *Bt* group larvae at 0 h, 6 h, 12 h and 24 h was counted, the mortality rate of each time period was calculated and the survival curve was analyzed. At the same time, the live insects of the CK group and *Bt* group were dissected at each time point to observe the appearance of midgut tissue.

### 4.4. Sample Collection

For the *Bt* group, three replicate groups were set up, each replicate group containing 30 larvae. Each larva was injected with an equal volume of *Bt* bacterial solution (LC_50_ = 3.3 × 10^8^ spores mL^−1^), and a total of 90 larvae were injected. The midgut tissues of live insects were sampled at 0 h, 6 h, 12 h and 24 h after *Bt* treatment. The PBS solution was placed in a petri dish containing ice, the larvae were rinsed for 20–30 s and the water was then dried on sterile filter paper. The surface of the third abdominal foot of the larva was cut off with ophthalmologists, the midgut tissue was taken out and the surface water was gently sucked with filter paper. For transmission electron microscope slice samples, after removal, they were quickly placed in 2 mL centrifuge tubes filled with glutaraldehyde solution (2.5%), fixed at 4 °C and stored overnight for transmission electron microscope slice preparation. For enzyme activity samples and metabolome samples, after removal, they were quickly placed in a 2 mL non-enzymatic centrifuge tube, temporarily stored in liquid nitrogen and then stored in an ultra-low-temperature refrigerator at −80 °C for later use. For transcriptome sequencing samples, the midgut tissue samples were immediately placed in an RNA stable preservation solution and stored at 4 °C. After 24 h, the RNA stable preservation solution was dried and stored in a −80 °C ultra-low-temperature refrigerator for later use.

### 4.5. MDA Content Change and Enzyme Activity Determination

The midgut tissue samples of larvae *Bt*0h, *Bt*6h, *Bt*12h and *Bt*24h were sent to Suzhou Grace Biotechnology Co., Ltd. (No.98 Jujin Road, Taiping Street, Xiangcheng District, Suzhou City, Jiangsu Province, China), and malondialdehyde (MDA) content, catalase (CAT) activity, superoxide dismutase (SOD) activity and glutathione peroxidase (GPx) activity were determined by kit method. The article numbers are as follows: G0109W, G0105W, G0101W and G0204W, respectively. Excel (2016) software is used for data statistics, IBMSPSS Statistics 24 is used for independent sample *T*-test analysis and Graphpad Prism 9.0 is used for graphic drawing.

### 4.6. Identification and Analysis of Midgut Metabolites

The extraction, separation and LC-MS analysis of metabolites in midgut tissues of *Bt*0h, *Bt*6h, *Bt*12h and *Bt*24h were completed by Shanghai Meiji Biomedical Technology Co., Ltd.(No.3377 Kangxin Highway, Pudong New Area, Shanghai, China). After the computing is finished, the LC-MS raw data is imported into the metabolomics processing software Progenesis QI 2.0 (Waters Corporation, Milford, CT, USA) for baseline filtering, peak identification, integration, retention time correction and peak alignment, and finally a data matrix of retention time, mass-to-charge ratio and peak intensity is obtained. At the same time, the MS and MSMS mass spectrometry information was matched with the metabolic public databases HMDB (https://www.hmdb.ca/) (accessed on 10 October 2022) and Metlin (https://metlin.scripps.edu/) (accessed on 20 October 2022) and Meggie’s self-built database, and the metabolite information was obtained. The searched data matrix was uploaded to the Meggie Cloud Platform (cloud.majorbio.com) (accessed on 25 October 2022) for analysis. Firstly, the data matrix is preprocessed to obtain the normalized data matrix. At the same time, the variables whose relative standard deviation (RSD) of QC samples is more than 30% are deleted, and the log10 is logarithmized to obtain the data matrix for subsequent analysis. The choice of significantly different metabolites was determined based on the variable weight (VIP) obtained by the OPLS-DA model and the *p* value of Student’s *t*-test. The metabolites with VIP > 1 and *p* < 0.05 were significantly different metabolites. Differential metabolites were annotated by the KEGG database (https://www.genome.jp/kegg/pathway.html) (accessed on 25 October 2022) to obtain the pathways in which differential metabolites participated. The Python software (accessed on 25 October 2022)package scipy.stats was used for pathway enrichment analysis, and the most relevant biological pathway was obtained through Fisher’s exact test.

### 4.7. Transcriptome Library Construction and Sequencing Data Analysis

Total RNA was extracted from the midgut tissue samples of *Bt*0h, *Bt*6h, *Bt*12h and *Bt*24h groups by the Trizol method. After the quality and integrity of total RNA were tested, four cDNA libraries were established by using the MGIEasy RNA Library Preparation Kit (Shenzhen Huada Zhizao Technology Co., Ltd.) (No.21, Hong ’an Third Street, Yantian District, Shenzhen, China). The qualified library was sequenced by the DNBSEQ-T7 (formerly MGISEQ-T7) sequencing platform. The library construction and sequencing were completed by Wuhan Hope Group Biotechnology Co., Ltd. (No.8 Huacheng Avenue, Hongshan District, Wuhan City, China). The original sequencing data were filtered by the software fastp [54] (https://github.com/OpenGene/fastp) (accessed on 30 October 2022). The clean data after quality control was compared with the reference genome of *D. kikuchii* (accession number: GCA_019925095.2) by the software HiSat2 [55] (http://daehwankimlab.github.io/hisat2/) (accessed on 2 November 2022). RSEM [56] (http://deweylab.github.io/RSEM/) (accessed on 5 November 2022) software was used to quantitatively analyze the expression levels of genes and transcripts, and the results of expression quantification were based on TPM. The homogenization process of TPM made the total expression in different samples consistent. DESeq2 software [57] (https://bioconductor.org/packages/stats/bioc/DESeq2/) (accessed on 6 November 2022) was used to analyze the expression differences of the sample genes that met the screening conditions of p-adjust < 0.05 and |log2FC| ≥ 1, and the number of differentially expressed genes (DEGs) among the samples was counted. KEGG annotation of differentially expressed genes was performed by the KEGG database (https://www.genome.jp/kegg/) (accessed on 6 November 2022)), and KEGG pathway enrichment analysis was performed by KOBAS (http://kobas.cbi.pku.edu.cn/home.do) (accessed on 10 November 2022) software [58].

### 4.8. Real-Time Fluorescence Quantitative PCR Verification

Real-time fluorescence quantitative PCR (RT-qPCR) with β-actin as reference gene was used to verify the accuracy of transcriptome sequencing results. RT-qRCR primers were designed by Primer Premier 6.0 software, which were synthesized by Beijing Qingke Biological Company (No.156 Jinghai 4th Road, Economic and Technological Development Zone, Tongzhou District, Beijing, China) (Table 2). The experimental data were analyzed by the 2^−△△Ct^ method, and GraphPad Prism 9.0 was used to draw.

### 4.9. Transcriptional Metabolic Correlation Analysis

Through the data association of the Meggie platform project, the transcriptome data and metabolomics data are integrated and analyzed by the method of generating letters so as to achieve the purpose of efficient and personalized data mining. Related biological problems from the level of metabolites and mRNA were compared and studied, and key molecules were further explored.

## 5. Conclusions

Like other insects, *Bt* also has toxic effects on the 5th instar larvae of *D. kikuchii*. After *Bt* infection, the midgut cells of the larvae were diseased, the structure and function of the organelles were damaged, the midgut tissue was lysed and the larvae died. At the same time, the transcriptome and metabolic groups of the 5th instar larvae of *D. kikuchii* after *Bt* infection changed. A total of 27 DEMs and 388 DEGs were annotated to seven signaling pathways, among which Nicotinate and nicotinamide metabolism, Lysine degradation, Longevity regulating pathway—worm and Vitamin digestion and absorption had higher influence weights. Nicotinate and nicotinamide metabolism signaling pathways and nicotinamide are the most important affected signaling pathways and metabolites. The expressions of two differential metabolites of nicotinamide and L-2-Aminoadipic acid were down-regulated. In addition to the up-regulated expression of the *SLC23A1* gene, the expression of *NNT*, *ALDH*, *SETMAR*, *GST*, *PNLIP* and *RNASEK* was down-regulated. It should be emphasized that they are all involved in the detoxification defense process in the body. It is speculated that these genes and metabolites are likely to be the key genes and important metabolites of *D. kikuchii* in response to *Bt* infection, and participate in the defense mechanisms of *D. kikuchii*.

## Figures and Tables

**Figure 1 ijms-25-11823-f001:**
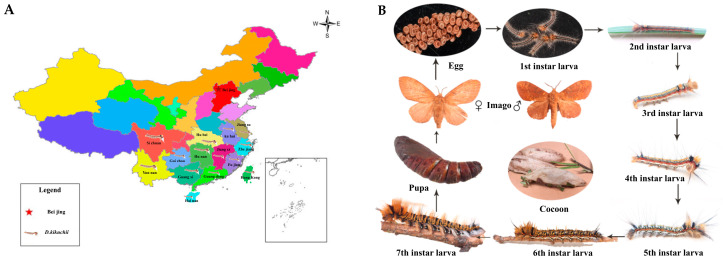
Introduction of *D. kikuchii*. (**A**) Distribution of *D. kikuchii* in China. (**B**) Life history of *D. kikuchii* [4].

**Figure 2 ijms-25-11823-f002:**
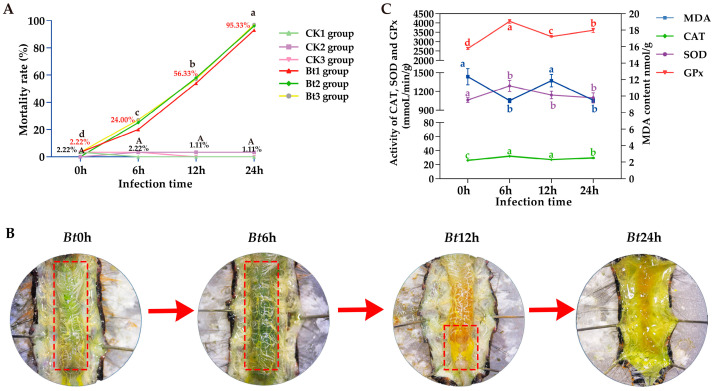
Toxic effects of *Bt* on the 5th instar larvae of *D. kikuchii*. (**A**) After eating pine needles soaked in PBS/*Bt* bacterial solution, the mortality rate of the 5th instar larvae of *D. kikuchii* was changed at CK, *Bt*0h, *Bt*6h, *Bt*12h and *Bt*24h. The mean value of the three biological repeated mortality rates of each comparison group is marked in the figure. The red font is the mean value of the mortality rate of each *Bt* infection time group, and the black font is the mean value of the mortality rate of each time group of the CK group. There was no significant difference in mortality between different time periods in CK group, but there was a significant difference in mortality between different infection time periods in *Bt* group, and almost all the larvae died after 24 h of infection. In the figure, capital letters indicate the difference of the average mortality of three biological repeats in CK group at different infection time nodes, and all of them are A, indicating that there is no difference in the mortality of each infection time node in CK group; Whether there is a significant difference between the average mortality of three biological repetitions at each infection time node in *Bt* group is indicated by lowercase letters. In the figure, the average mortality of 24 h is the highest, marked as a, followed by 12 h, 6 h and 0 h, and marked as b, c and d in turn because there are significant differences between the two. (**B**) The changes of the midgut tissue of the 5th instar larvae of *D. kikuchii* after eating the pine needles soaked in PBS/*Bt* bacterial solution. In *Bt*0h, the midgut tissue structure of larvae was complete and light green. For *Bt*6h, it can be observed that the midgut tissue structure is still intact, but the color becomes dark green; in *Bt*12h, the color of the first two groups has changed significantly, from dark green to yellow, and the lower part of the midgut has been festered and liquefied. By 24 h after *Bt* infection, the midgut tissue structure of the larvae had completely lost its complete structure and was completely liquefied. (**C**) The 5th instar of *D. kikuchii* after *Bt* infection. The a, b, c and d in the figure indicate that there are differences between groups.

**Figure 3 ijms-25-11823-f003:**
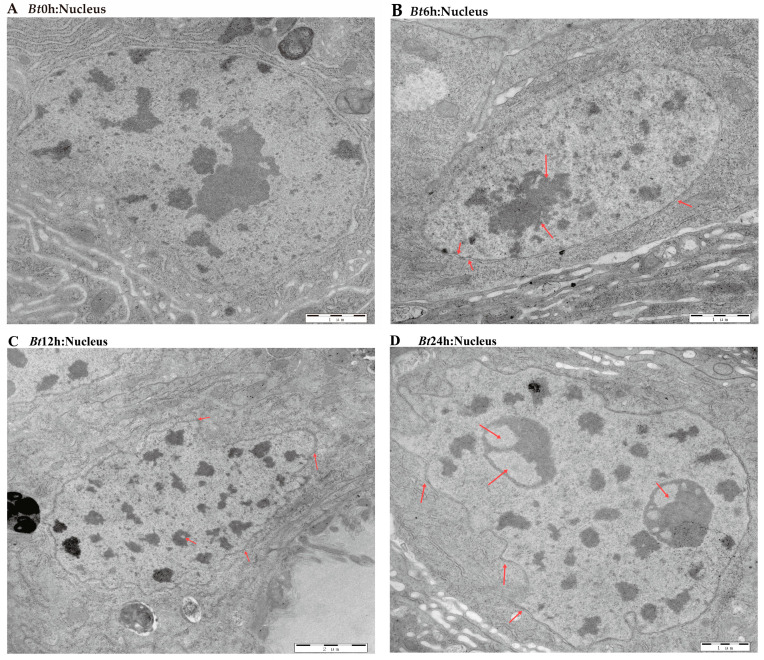
Pathological changes in cell nucleus in midgut tissue. (**A**) In the figure, the cell nucleus of the midgut tissue of the *Bt*0h group was full, the nuclear membrane structure was complete and smooth and the nucleoplasm was evenly distributed. (**B**) The red arrow in the figure indicates the nuclear membrane and nucleus of the lesion. Nucleolus aggregated, and the inner and outer membrane spaces of the nucleus were separated. (**C**) As shown by the red arrow, the nuclear lesion is aggravated, the nucleoli is hollow, the nuclear morphology is severely deformed, and the nuclear membrane is angular. (**D**) As shown by the red arrow, the nucleolus cavity becomes larger, the nucleus is seriously deformed and the nuclear membrane is obviously angular.

**Figure 4 ijms-25-11823-f004:**
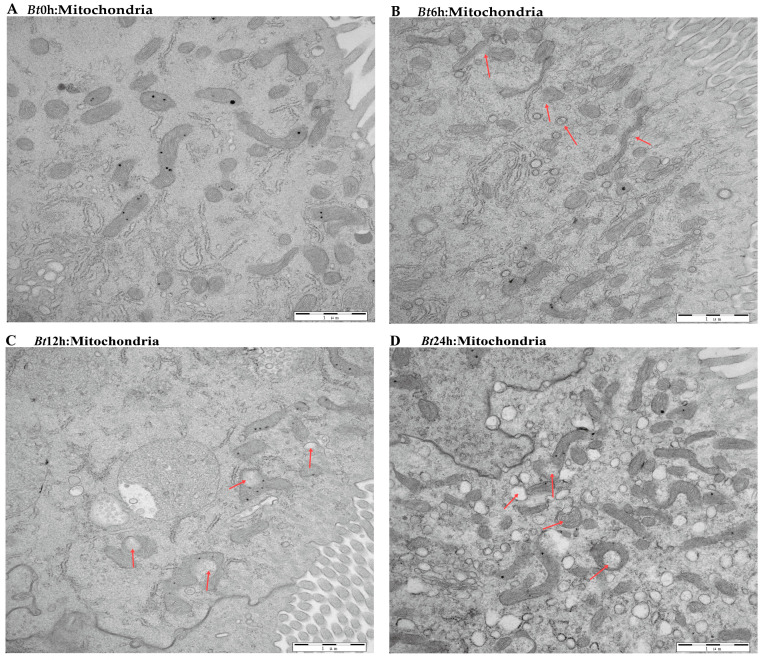
Mitochondrial pathological changes in midgut tissue cells. (**A**) In the figure, there are normal mitochondria, most of which are oval or fusiform, with a clear inner ridge and complete structure. (**B**) As shown by the red arrow, the mitochondrial morphology was deformed, the individual cells became smaller, most of the mitochondrial inner ridges were damaged or disappeared and the internal structure was blurred. (**C**) As shown by the red arrow, the mitochondria swelled, the internal structure degraded to form voids and the number decreased. (**D**) As shown by the red arrow, the mitochondrial morphology was severely deformed, and the cavity enlargement led to the formation of a large number of vesicles and serious degradation.

**Figure 5 ijms-25-11823-f005:**
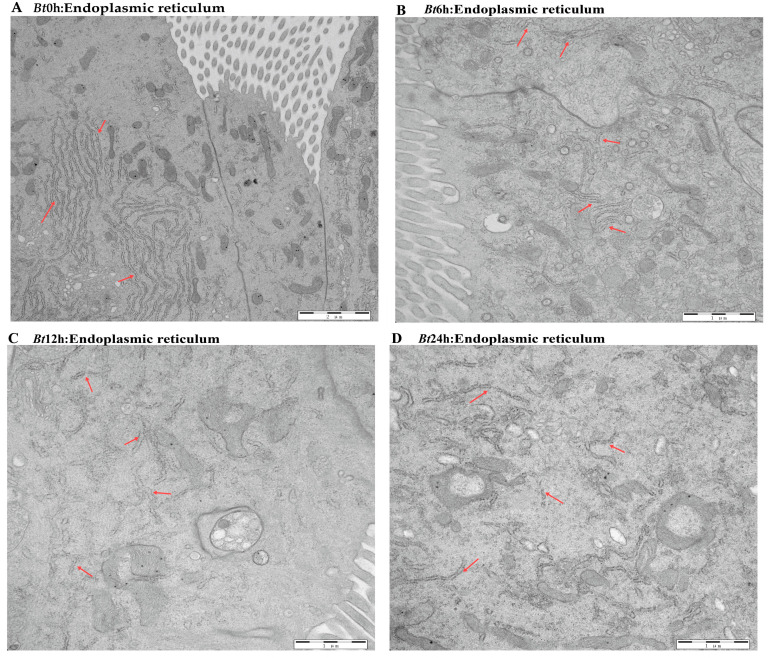
Pathological changes in endoplasmic reticulum in midgut tissue. (**A**) As shown by the red arrow, the endoplasmic reticulum in the *Bt*0h group was neatly arranged, with a large number and a complete structure. (**B**) As shown by the red arrow, the number of endoplasmic reticula decreased and the arrangement was disordered. (**C**,**D**) As shown by the red arrow, the endoplasmic reticulum is severely degraded, and only a very small number of undegraded endoplasmic reticula can be observed.

**Figure 6 ijms-25-11823-f006:**
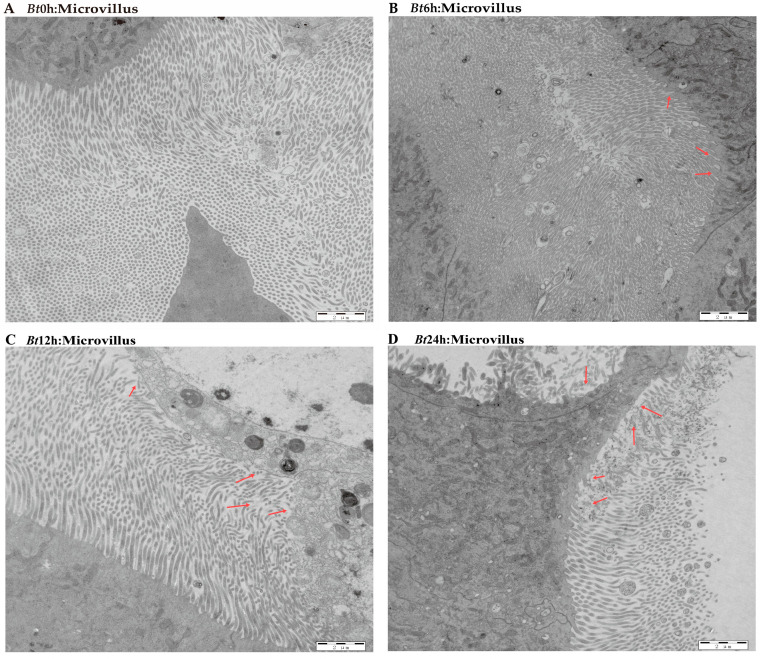
Pathological changes of microvilli in midgut tissue. (**A**) The figure shows the normal microvilli morphology, with a large number and neat arrangement. (**B**) The arrow indicates slightly swollen microvilli. (**C**) The arrangement of microvilli was disordered, the degree of swelling and deformation was aggravated and the shedding was serious. (**D**) The degree of deformation and shedding of microvilli is extremely serious and is degraded.

**Figure 7 ijms-25-11823-f007:**
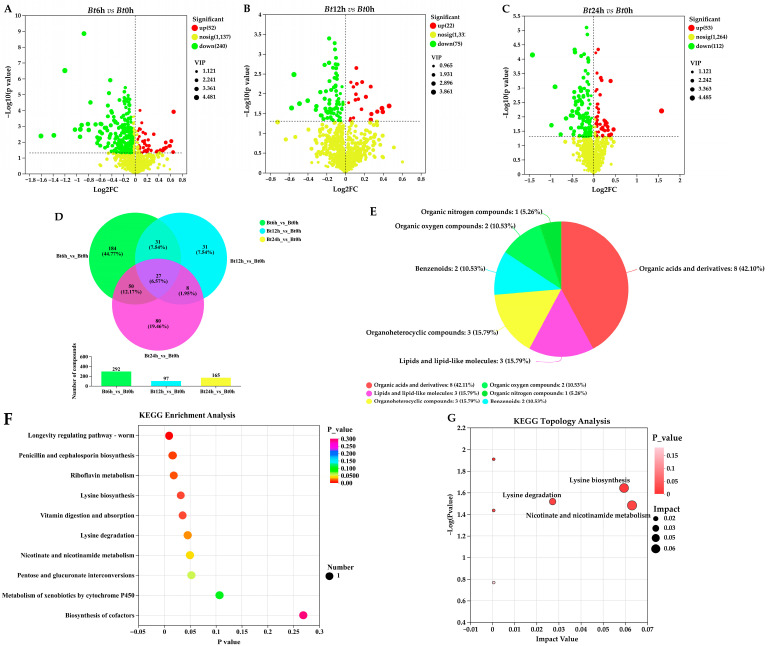
Analysis of different metabolites in midgut tissue. (**A**) Differential volcanic map of *Bt*6h vs. *Bt*0h formation. (**B**) Differential volcanic map of *Bt*12h vs. *Bt*0h formation. (**C**) Differential volcanic map of *Bt*24h vs. *Bt*0h formation. (**D**) Venn analysis of three groups of differential metabolites, with 27 intersecting differential metabolites obtained. The purple part is the intersection difference metabolite obtained. (**E**) Matching with the database of human metabolites, 27 different metabolites were classified. (**F**) KEGG enrichment analysis of 27 different metabolites showed that *p* < 0.05 was significant enrichment. (**G**) Based on the results of KEGG enrichment analysis, the topological analysis of the obtained signal pathways was carried out, among which the Nicotinate and nicotinamide signal pathways and the differential metabolite nicotinamide had the highest weight.

**Figure 8 ijms-25-11823-f008:**
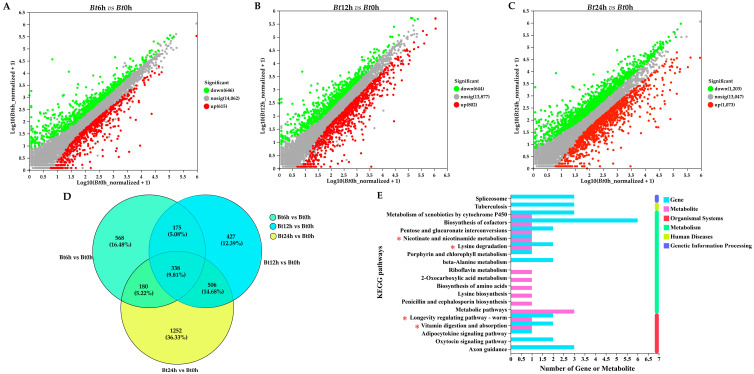
Transcriptome analysis of midgut tissue. (**A**) Volcano map of differentially expressed genes in *Bt*6h vs. *Bt*0h group. (**B**) Volcano map of differentially expressed genes in *Bt*12h vs. *Bt*0h group. (**C**) Volcano map of differentially expressed genes of *Bt*24h vs. *Bt*0h group. (**D**) Venn analysis of differentially expressed genes in 3 groups; 388 intersection differentially expressed genes were obtained. (**E**) KEGG annotation analysis of 388 differentially expressed genes. In the figure, the * marked pathway is the common signal pathway of transcription group and metabolism group.

**Figure 9 ijms-25-11823-f009:**
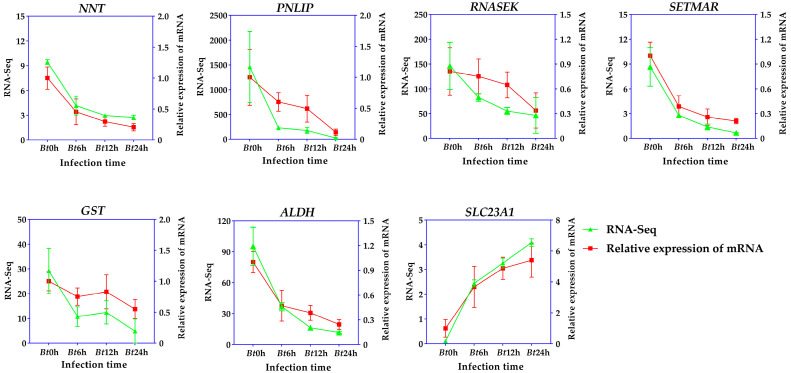
RT-qPCR verification.

**Table 1 ijms-25-11823-t001:** Four signal pathways, differentially expressed genes and differentiated metabolites.

KEGG Pathway ID	Signaling Pathway	DEGs	DEMs
map00760	Nicotinate and nicotinamide metabolism	NNT	Niacinamide
map00310	Lysine degradation	ALDH SETMAR	L-2-Aminoadipic acid
map04212	Longevity regulating pathway—worm	GST RNASEK	Niacinamide
map04977	Vitamin digestion and absorption	PNLIP SLC23A1	Niacinamide

**Table 2 ijms-25-11823-t002:** Gene primer information for RT-qPCR.

Gene ID	Primer Symbol	Sequence (5′-3′)
*D. kikuchii*_LG29_G00053	β-actin	F: GTAGGCACGAACAACAGA R: CAGAAGGTGATGTCAGGAG
*D. kikuchii*_LG16_G00317	ALDH	F: GGCACCGACAGACAGACACAG R: CCAGCAACACGCCTAGTCTATCC
*D. kikuchii*_LG12_G00198	SETMAR	F: TGTCACGGTCGCCAGATGTTAC R: ACGCTCCGAACGCAACCAG
*D. kikuchii*_LG05_G00429	NNT	F: TTCCGACATTAGTAGCCAGGTTCC R: GGAGTACAGCAGACAGGCAGAC
*D. kikuchii*_LG15_G00353	PNLIP	F: AGAGCGGCAGCACATTGGTAG R: ACACGAGTTCAGCAGGCAGAC
*D. kikuchii*_LG12_G00593	SLC23A1	F: GGAGGCGTTGGCGGCTTAG R: GGATGATGGTTGGTCGGTGGTAG
*D. kikuchii*_LG18_G00183	RNASEK	F: AGCTGTCTTGGTTGCCCTAATCC R: TGATGAGCCACACACTCTCGATC
*D. kikuchii*_LG07_G00366	GST-F	F: TAACGGCAACATCGTATCG R: TATGCCTGCGTACACTACT

## Data Availability

The data will be available on request.

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
