# Peer review of "Combined Analysis of Metabolomics and Transcriptome Revealed the Effect of *Bacillus thuringiensis* on the 5th Instar Larvae of *Dendrolimus kikuchii Matsumura"

_ijms, 2024, doi:10.3390/ijms252111823_

Round 1

Reviewer 1 Report

Comments and Suggestions for Authors

The present study aimed to explain the toxic effect of Bacillus thuringiensis (Bt) on the 5th instar larvae of D.kikuchii by analyzing the mortality, the anatomical observation of midgut tissue and cell biochemistry, pathological characteristics, and the metabonomics and transcriptomics methods were used for further analysis. The authors found that feeding on pine needles soaked in Bt defence the larval vitality, feeding and crawling behavior and cause larval mortality. Forth more, the midgut tissue of larvae showed seriously damaged with Bt treatments. This work shows a comprehensive and meticulous research, and the results also have interesting scientific significance. I have some questions asfollowing:

1. Figure 1B showed the toxic effects of Bt on midgut tissue of the 5th instar larvae of D.kikuchii. The image does not show a complete midgut, but rather some segment of midgut tissue. The author should clarify which specific location of the midgut it is? Suggest adding a schematic diagram or complete midgut tissue to indicate the specific location of the picture.

2. As described in the article in line 93-96” we observed the changes of vitality….” “there was no significant change in larval activity after feeding Bt for 6 hours…. the vitality was obviously weakened, and the feeding and crawling behaviors were not obvious” . Larvae vitality, feeding and crawling behaviors, these indicators are typical insect behavioral analysis that require observation data or video statistics to draw specific conclusions. Here, the author did not provide any relevant data support, only simple words descriptions, so the results do not have scientific validity. I suggest the author to supplement relevant experimental data to support this conclusion.

3. Due to the lack of behavioral experiments, it is recommended to rewrite the abstract and discussion of the paper after supplementation experiments.

4. The authors carried out the association analysis of metabolomics and transcriptome and found many metabolic signal pathways associated with Bt treatments on the 5th instar larvae, which lays a foundation for the research of Bt as a new biological control agent for D.kikuchii. The author conducted routine transcriptome analysis, lacking of conclusive analysis.

Author Response

Comments 1:Figure 1B showed the toxic effects of Bt on midgut tissue of the 5th instar larvae of D.kikuchii. The image does not show a complete midgut, but rather some segment of midgut tissue. The author should clarify which specific location of the midgut it is? Suggest adding a schematic diagram or complete midgut tissue to indicate the specific location of the picture.

Response 1: Thank you for your valuable comments on the manuscript. We have revised the picture, replaced it with a complete anatomical map of the midgut, and marked the midgut with a red dotted box.

Comments 2:As described in the article in line 93-96” we observed the changes of vitality….” “there was no significant change in larval activity after feeding Bt for 6 hours…. the vitality was obviously weakened, and the feeding and crawling behaviors were not obvious” . Larvae vitality, feeding and crawling behaviors, these indicators are typical insect behavioral analysis that require observation data or video statistics to draw specific conclusions. Here, the author did not provide any relevant data support, only simple words descriptions, so the results do not have scientific validity. I suggest the author to supplement relevant experimental data to support this conclusion.

Response 2: I feel sorry for what we designed in this experiment, because we did observe the decline of larvae' vitality, feeding and crawling behavior with the extension of the experiment, but it's a pity that we didn't use the correct research method of insect behavior to make scientific records. Therefore, in order to avoid making wrong conclusions about the whole experiment and the results of the article, I will delete the elaboration on larval vitality, feeding and crawling behavior in the main body.

Comments 3:Due to the lack of behavioral experiments, it is recommended to rewrite the abstract and discussion of the paper after supplementation experiments.

Response 3:Consistent with the answer to the second question, in order to avoid making wrong conclusions about the whole experiment and the results of the article, I will delete the elaboration on larval vitality, feeding and crawling behavior in the main body.

Reviewer 2 Report

Comments and Suggestions for Authors

A well-written paper on biocontrol of one of the pests Dendrolimus kikuchii Matsumura. The authors analyzed the mortality of larvae after eating Bt-treated pine needles, observed the midgut tissues and cell biochemistry. The mortality of larvae reached 95.33% after 24 h. The midgut tissue of larvae was severely damaged with the increase of feeding time. Associative analysis of metabolomics and transcriptome showed that many metabolic signaling pathways, metabolites and gene expression levels in larvae were changed. This study was prepared properly. Figures are clear and present the obtained results.

The microscopic photos show the changed tissues well. How BT affects the feeding and development of harmful insects could be enriched with statistical analysis in future studies. References include 58 references.

Author Response

Comment 1: A well-written paper on biocontrol of one of the pests Dendrolimus kikuchii Matsumura. The authors analyzed the mortality of larvae after eating Bt-treated pine needles, observed the midgut tissues and cell biochemistry. The mortality of larvae reached 95.33% after 24 h. The midgut tissue of larvae was severely damaged with the increase of feeding time. Associative analysis of metabolomics and transcriptome showed that many metabolic signaling pathways, metabolites and gene expression levels in larvae were changed. This study was prepared properly. Figures are clear and present the obtained results.

Response 1: Thank you for your recognition of this experimental work and your hard work in reviewing the manuscript.

Comment 2: The microscopic photos show the changed tissues well. How BT affects the feeding and development of harmful insects could be enriched with statistical analysis in future studies. References include 58 references.

Response 2: Once again, I sincerely thank you for your recognition of this experimental work and your hard work in reviewing the manuscript. I give you my best wishes.